# Methanogenic Archaea in the Pediatric Inflammatory Bowel Disease in Relation to Disease Type and Activity

**DOI:** 10.3390/ijms25010673

**Published:** 2024-01-04

**Authors:** Agata Anna Cisek, Edyta Szymańska, Aldona Wierzbicka-Rucińska, Tamara Aleksandrzak-Piekarczyk, Bożena Cukrowska

**Affiliations:** 1Department of Pathomorphology, The Children’s Memorial Health Institute, Av. Dzieci Polskich 20, 04-730 Warsaw, Poland; gutkac@op.pl; 2Department of Gastroenterology, Hepatology, Nutritional Disorders and Pediatrics, The Children’s Memorial Health Institute, Av. Dzieci Polskich 20, 04-730 Warsaw, Poland; edyta.szymanska@ipczd.pl; 3Department of Biochemistry, Radioimmunology and Experimental Medicine, The Children’s Memorial Health Institute, Av. Dzieci Polskich 20, 04-730 Warsaw, Poland; a.wierzbicka-rucinska@ipczd.pl; 4Institute of Biochemistry and Biophysics, Polish Academy of Sciences, Pawińskiego 5a, 02-106 Warsaw, Poland; tamara@ibb.waw.pl

**Keywords:** Crohn’s disease, ulcerative colitis, inflammatory bowel disease, children, fecal calprotectin, methanogens, archaea

## Abstract

The inflammatory bowel disease (IBD) is associated with gut microbiota dysbiosis; however, studies on methanogens—especially those focused on children—are extremely limited. The aim of this study was to determine the abundance of total methanogenic archaea and their three subgroups: *Methanobrevibacter* (*Mb.*) *smithii*, *Methanosphaera* (*Ms.*) *stadtmanae*, and Methanomassiliicoccales, in the feces of children with both active and inactive Crohn’s disease (CD) and ulcerative colitis (UC). The results of a quantitative real-time PCR were cross-referenced with the disease type (CD vs. UC) and activity assessed with the use of Pediatric Crohn’s Disease Activity Index (PCDAI) and Pediatric Ulcerative Colitis Activity Index (PUCAI) indices, and fecal calprotectin (FCP) concentration, and compared with controls. There was a significant decrease in the number of total methanogens in CD and UC compared to controls. The prevalence of total methanogens was also lower in UC compared to controls. Furthermore, patients from the inactive UC group were colonized by a lower number of *Mb. smithii*, and demonstrated the most pronounced positive correlation between the number of *Ms. stadtmanae* and the FCP concentration. Our results demonstrate that gut methanogens are related to the type and activity of pediatric IBD.

## 1. Introduction

Inflammatory bowel disease (IBD) is a collective name for two disease conditions—Crohn’s disease (CD) and ulcerative colitis (UC)—which manifest in chronic inflammation of the gastrointestinal tract and usually result in recurrent diarrhea, abdominal pain, loss of appetite, and weight loss [1]. In some cases, the consequences of IBD may be severe. Over time, IBD can lead to ulcers, fistulas, bowel obstruction, and sepsis [1]. It is estimated that IBD affects up to 0.3% of people living in developed countries [2], with a quarter of new cases diagnosed in children under 18 years of age [3]. The occurrence of IBD varies geographically. The highest incidence is observed in developed countries of Western Europe (40–50 cases in every 100,000 inhabitants per year) and North America (3.1–14.6 cases in every 100,000 inhabitants per year) [4]. IBD occurs at any age, but the peak incidence occurs in the 2nd–3rd decade of life, with 15% of IBD cases affecting children [5]. In Poland, the annual detection rate is 2.8 cases in every 100,000 children up to 15 years of age [3]. Etiopathology of CD and UC is still unknown and most probably multifactorial, including genetic background, immunological dysfunction, dysbiosis, and socio-demographic factors such as diet, place of residence, etc. [6,7].

It has been proven that intestinal bacterial dysbiosis is one of the causes of the initiation and progression of IBD [8]. However, there are some indications that the changes include not only bacteria but also methanogenic archaea as well [9]. For a long time, archaea were known only as peculiar, single-celled colonizers of extreme environments [10]. That changed in 1968 when methanogens—a group of archaea—were isolated from the human intestines [11]. Since then, methanogenic archaea have been considered a normal part of the gut microbiota [12]. In healthy individuals, this group of archaea constitutes up to 10% of all anaerobes present in the intestines [13]. Among them, three taxa are of the greatest importance, namely *Methanobrevibacter* (*Mb.*) *smithii*, *Methanosphaera* (*Ms.*) *stadtmanae*, and methanogens from the order Methanomassiliicoccales [14]. The impact of these three taxa on human health is diverse: in adults, *Mb. smithii* appears to be a commensal species [15], while *Ms. stadtmanae* strongly induces an immune response both in healthy people and in patients with IBD [13,16,17].

In adults with IBD, there is a reduced number of total methanogenic archaea and a reduced number of *Mb. smithii* were observed [9]. Moreover, also in adults with IBD, an increase in the share of *Ms. stadtmanae* was reported [16]. In children, none of the relationships described above have been confirmed [18,19]. Since pediatric IBD is significantly different from the one observed in adults in terms of progression, anatomical location, and treatment results [20], it is extremely important to understand the mechanisms behind the development of this disease in both groups separately. Moreover, children seem to be the best model for research on the pathomechanisms of IBD because they are rarely affected by other diseases, making it possible to learn the real causes underlying the initiation and development of IBD [8]. For these reasons, the aim of this study was to determine the abundance of methanogenic archaea in the feces of children with Crohn’s disease (CD) and ulcerative colitis (UC) and to estimate the relationship of the type of the disease (CD versus UC), and the activity of the IBD (active vs. inactive) compared to controls.

## 2. Results

### 2.1. Characteristics of the Subjects

The study comprised 97 children with IBD, including 45 with CD at a mean age of 14.2 years and 52 with UC at a mean age of 13.0 years (Table 1). Children were divided into groups based on the disease activity: the active and inactive ones. The disease activity was established based on the fecal calprotectin (FCP) concentration and diseases’ activity indices, i.e., the Pediatric Crohn’s Disease Activity Index (PCDAI) and the Pediatric Ulcerative Colitis Activity Index (PUCAI). No statistically significant differences were found when assessing the age and gender of children between the IBD groups. The mean PCDAI and PUCAI scores were 10.0 (±13.5) and 15.2 (±20.3), respectively, and were statistically significantly higher (*p* = 0.0027 and *p* = 0.0017) in the group with the active type of the disease compared to the inactive type. The mean FCP concentration in the CD and UC groups was 781.6 (±1800.7) and 396.6 (±810.3) µg/g, respectively, and was statistically significantly higher in both groups with the active type of the disease (i.e., active CD and active UC) compared to the inactive types (*p* < 0.0001 and *p* = 0.0002) and the control group (*p* < 0.0001), where the mean concentration of FCP was 19.3 (±24.1) µg/g.

No statistically significant differences in age were found between children with the active UC form compared to controls (*p* > 0.05).

### 2.2. The Prevalence and Abundance of Gut Methanogens

In the first step, a qualitative and quantitative analysis of methanogens was performed in the children’s stool samples. In contrast to the control group, where methanogens were found in all tested children, the IBD group was characterized by a substantial percentage of children in whom the presence of methanogens was not found (Table 2). The prevalence of total methanogens was statistically significantly lower in the group of children with UC and active UC (*p* < 0.05). Moreover, the analysis of specific groups of methanogens showed that *Ms. stadtmanae* was significantly less frequently present in children with UC compared to controls (*p* < 0.05). Furthermore, the odds of detecting each subgroup of methanogens were smaller in most of the IBD groups (except for the active CD patients) compared to controls (Table 3).

The average abundance of total methanogens was determined at 3.66, 2.87, and 5.04 log_10_/g of dry weight in patients with CD, UC, and controls, respectively, whereas the mean medians were 3.55, 3.07, and 4.30 log_10_/g of dry weight in CD, UC, and controls, respectively. There was a statistically significant difference in the number of total methanogenic archaea between both CD and UC patients and controls (*p* < 0.05) (Figure 1A). Interestingly, a significant difference was also observed between patients with CD and UC. In contrast, when the three subgroups of methanogens, i.e., *Mb. smithii*, *Ms. stadtmanae,* and Methanomassiliicoccales were tested individually, only the levels of *Mb. smithii* differed significantly between the UC and the control group (*p* = 0.0015) (Figure 1B). The differences among the other two taxa, i.e., Methanomassiliicoccales and *Ms. stadtmanae*, were not statistically proven (Figure 1C,D).

A more detailed analysis was performed with respect to the activity of the disease in the IBD groups, and it yielded similar observations in terms of UC vs. control groups. Both forms of UC, i.e., the active and the inactive one, demonstrated far lower total methanogen quantities than the control group (*p* = 0.0002 and *p* < 0.0001) (Figure 2A). However, in CD patients, the results were quite different—neither the active nor the inactive CD groups demonstrated statistically significant differences in terms of total methanogen quantities when compared to the control group. As for *Mb. smithii*, the statistical importance was observed only between the inactive UC patients and the control groups (*p* = 0.002) (Figure 2B). The differences in numbers of Methanomassiliicoccales and *Ms. stadtmanae* remained not statistically proven (Figure 2C,D).

### 2.3. Relationship between IBD Activity Indices and the Number of Methanogenic Archaea

The analysis of associations between CD and UC activity indices (PCDAI and PUCAI) and the abundance of methanogenic archaea revealed only one moderate positive correlation between the total methanogen counts and PCDAI values in the active form of CD (Rs 0.48, *p* = 0.026) (Figure 3). The remaining archaeal taxa proved not to be correlated with the disease activity indices.

### 2.4. Relationship between FCP and the Number of Methanogenic Archaea

It is known that FCP level correlates with the severity of IBD; thus, we decided to analyze the relation between FCP concentration and the abundance of total methanogens, *Mb. smithii*, *Ms. stadtamanae*, and Methanomassillicoccales. The analysis of the individual disease activity group revealed only one moderately positive correlation between the *Ms. stadtmanae* load and the inactive UC group of patients (Rs = 0.41, *p* = 0.034) (Figure 4).

### 2.5. Relationship between Age and the Number of Methanogenic Archaea

Due to significant age differences between the control group and children with IBD (Table 1), we decided to statistically analyze the presence of methanogens depending on age. The average age of the patients was 13.9 (±3.4) in the active CD group, 14.6 (±2.8) in the inactive CD group, 12.2 (±5.0) in the active UC group, 13.8 (±4.3) in the inactive UC group, and 10.0 (±4.0) in controls. The Spearman’s rank test showed that the only statistically significant association among the IBD groups was observed in the active CD patients, where, in contrast to controls, total methanogen counts and *Mb. smithii* quantities decreased with the patient’s age (Rs = –0.56, *p* = 0.009 and Rs = –0.53, *p* = 0.013, respectively). On the other hand, there has been a significant increase (Rs = 0.49, *p* = 0.009) in the number of Methanomassiliicoccales in association with age in the control group (Figure 5).

## 3. Discussion

Dysbiosis of the intestinal microbiome seems to play a significant role in the mechanisms leading to the development of IBD, but only a few studies devote their attention to the importance of methanogens in this process, the vast majority of which concern adult patients [9,16,23,24,25]. 

To date, only one team has analyzed in detail the intestinal methanogens in children with CD [19]. To the authors’ knowledge, the current research is the first to evaluate the prevalence of methanogens in UC separately from CD and the first to analyze the association between the presence of as many as four groups of methanogens and the activity of both types of IBD. 

Our results demonstrated a significant drop in the abundance of total methanogens in both IBD groups compared to controls, and this observation was the only common feature of CD and UC. This outcome is not surprising, given that both conditions demonstrate very different pathomechanisms and the course of disease [3]. 

Patients with UC were colonized by methanogens not only with a lower abundance but also less frequently compared to controls. Moreover, they were characterized by decreased quantities of lower methanogenic taxa, such as *Mb. smithii*. Based on the fact that reduced *Mb. smithii* population is a common signature of gut microbiota dysbiosis in IBD; our study provided yet another evidence supporting this phenomenon [23,26,27]. 

The literature data on another methanogenic species, *Ms. stadtmanae*, indicated that it can strongly induce some inflammatory responses not only in the IBD-affected gut but also in the healthy intestine, much more than any other methanogens tested [16,28]. Our data did not seem to support this, but we were able to provide another interesting fact compatible with this theory. We observed a statistically important, positive correlation (only in the inactive form of UC) between the levels of *Ms. stadtmanae* and FCP, a known indicator of the inflammatory processes hollowing the gut mucosa, and a biomarker of IBD severity [29]. Moreover, a similar correlation (albeit not statistically proven) was observed in all four methanogenic groups in control patients. Therefore, perhaps all intestinal methanogens contribute to the induction or persistence of inflammatory responses, some more than others. 

In contrast to UC, patients with CD had similar prevalence indices of all four methanogenic groups tested as the control patients. Our results align with the study by Krawczyk et al., 2021 [19]. However, the authors reported that *Mb. smithii* was present in 44% of active CD cases, 27% of inactive CD cases, and in 36% of controls. In our study, these prevalence values were higher, as 76% of the active CD patients, 63% of the inactive CD patients, and 74% of the controls tested positive for *Mb. smithii*. The percentages for *Ms. stadtmanae* also differed: in our study, these were 38%, 17%, and 37% in the active CD, inactive CD, and controls, respectively, whereas in another study, they were all around 30% [19]. In adults, *Ms. stadtmanae* is three times more frequent in IBD patients [16], which was not confirmed in our study, as only 27% and 15% of children in the CD and UC group, respectively, were colonized by this archaeal species compared to 37% of the controls.

There are some reports in adult patients with IBD that are, to some extent, reflected in our results, and we recommend that any future observations and findings made on children be compared with those of the adults. In adults, decreased methane production has already been well-documented in IBD cases [30,31,32,33]. Based on the fact that there must be at least 8 log_10_/g of dry weight of feces in order to detect methane in breath [34], in our study, all children in the active IBD groups would probably not be methane producers. On the other hand, some children in the control group would likely be methane producers—specifically, 2 of the 15 (13%) children aged 8 to 14 years and 2 out of the 5 (40%) children aged 14 to 18 years. It is consistent with a study by Peled et al., 1985 [35], who reported that approx. 14 to 18% and 9 to 46% of children aged 7 to 14, and 14 to 18, respectively, were methane producers.

Since the control group was age-matched only with the active UC patients, we decided to analyze the relationship between age and methanogen counts in all four IBD groups. Interestingly, we noticed a significant drop in the total number of methanogens and *Mb. smithii* with the age of active CD patients. A similar tendency (albeit not statistically proven) for decline was observed in all other groups of IBD. Moreover, a positive correlation was observed between the PCDAI indices and the number of total methanogens in active CD patients. It is not clear whether these results were a cause or a consequence of the CD, but we speculate that perhaps methanogens may somehow initiate the disease (judging by the correlation between the PCDAI vs. total methanogen counts). As the disease persists for years, especially in its active form, the methanogen population subsequently declines (based on age vs. total methanogen counts correlation). Surely, this hypothesis requires further investigation [19]. 

In terms of the age-related differences, one additional observation was made. The control group was characterized by a pronounced rise in Methanomassiliicoccales as the children’s age increased. This tendency is well-documented in adults and children [36,37], and we have provided more evidence of this phenomenon here. Moreover, Vanderhaeghen et al. 2015 reported that in single cases, Methanomassiliicoccales could dominate over Methanobacteriales (most of them belong to *Mb. smithii*) [38], which held true in our study, and, importantly, was not restricted to a single group of patients.

### Limitation and Strength of the Study

We realize that the results should be interpreted cautiously as the research encountered limitations. This study was performed on a small cohort of patients; not all cases could be statistically determined. The age of the control group was statistically lower than that of patients with active and inactive CD and inactive UC, which may have influenced some of the results. Moreover, the usage of therapeutics and their possible impact on the results cannot be ruled out. Patients were treated with various agents, which were divided into five groups such as 5-ASA agents (mesalazine), immunosuppressants (azathioprine), biological drugs (infliximab, adalimumab, vedolizumab, and ustekinumab), steroids (prednisone and budesonide), and nutritional therapy. There is some evidence in the literature suggesting that methanogens may be affected by therapeutics used in IBD. For instance, the use of mesalazine has been shown to increase or decrease the amount of some methanogens [16,23,27]. No such correlation was observed in our study. This was probably due to the small number of observations per therapy, and it is possible that some correlations may have been missed. Importantly, even though there were methanogen-negative samples, they were never grouped solely into a particular drug group. Therefore, none of the prescribed therapeutic agents could have been associated with a complete elimination of methanogens. On the other hand, low *Mb. smithii* quantities have been successfully used as an indicator to measure the probability of disease response to anti-TNF drugs (biological therapy of IBD) [39], which is yet another interesting topic that requires further investigation. Furthermore, it is possible that our pediatric patients might have been colonized by other unstudied taxa, which we did not study, and it cannot be ruled out that these taxa might have been somehow related to the incidence of IBD in these patients. This conclusion is due to the fact that the most pronounced differences between the patient groups were observed in quantities of total methanogens but not so much within the subgroups of methanogens. This issue will remain open until we learn more about the diversity of methanogenic archaea in the children’s intestines.

On the other hand, to the best of our knowledge, this study is the only one that evaluates the prevalence and abundance of four groups of methanogens in pediatric UC and CD separately from each other. This study is also the first to cross-reference methanogens with FCP and PUCAI in UC.

## 4. Materials and Methods

### 4.1. Subjects 

A total of 124 children, including the IBD patients (n = 97) and controls (n = 27), were recruited in this study. All of the patients were admitted into the Department of Gastroenterology, Hepatology, Nutritional Disorders and Pediatrics, the Children’s Memorial Health Institute (Warsaw, Poland), and their disease activity was established according to European Crohn’s and Colitis Organization (ECCO) [40]. All of them were between 3 and 18 years old. There were 45 patients with CD and 52 patients with UC. The control group consisted of 27 children reported for fecal examination due to non-IBD-related illnesses (Appendix A). The group of IBD patients was further divided into children with active and inactive CD (21 and 24 patients, respectively) and children with active and inactive UC (25 and 27 patients, respectively).

The division into subgroups depending on disease activity was performed based on PCDAI or PUCAI and the FCP concentration in feces. A PCDAI under 12.5 points indicated an inactive form of CD, whereas a PCDAI above 20 is an active form of the disease. In addition, progression or remission of CD with time was taken into account, especially in three children with PCDAI between 12.5 and 20, who were finally subjected to the active CD group. Patients with UC whose PUCAI was under 10 were considered being in remission, those with PUCAI between 10 and 30 had a mild form of UC, and those above 30—suffered from exacerbation of UC. For that matter, the term inactive UC refers to both mild forms of UC and UC in remission. In addition, patients with low activity indices and high calprotectin levels were assigned to the active groups of IBD. FCP above 200 µg/g was considered a high concentration.

### 4.2. FCP Measurement

The FCP concentration was measured using magnetic microparticle chemiluminescence technology in Liaison^®^ XL (DiaSorin, Saluggia, Italy). The assay uses a mouse monoclonal antibody on particles to capture FCP from stool samples and a second conjugated mouse monoclonal antibody against a different region of FCP for detection. A total of 15 mg of freshly extracted feces was collected using a serrated stick in a standard volume of extraction buffer. The range of reported measurements was 5 to 800 μg FCP/g of stool. Samples with high FCP concentrations were appropriately diluted, and the results were then multiplied by the dilution factor.

### 4.3. DNA Isolation

A total of 100 mg of each stool sample was weighed out into 2 mL microtubes and subjected to DNA isolation. If the sample had a high water-to-fecal content ratio, the weight was increased up to 300 mg, which was taken into account in the final calculations. The following DNA isolation procedure was partially described elsewhere [41], with the main difference concerning a mechanical lysis step, which was here improved in terms of hands-on time by replacing sonication with bead-beating. In brief, feces were suspended in the BS buffer until becoming viscous (A&A Biotechnology, Gdynia, Poland). After adding 30 µL lysozyme (10 mg/mL) and 7 µL mutanolysin (10 U/µL), the samples were incubated at 37 °C for 15 min and then at 50 °C for 25 min. Approx. 700 µL LS lysis buffer (A&A Biotechnology, Gdynia, Poland) and 35 µL proteinase K (20 mg/µL) were added, and the samples were incubated at 50 °C for 1 h. The samples were then centrifuged at 14,000 rpm for 5 min. The supernatant was collected into a separate microtube, whilst the zirconia/silica beads (A&A Biotechnology, Gdynia, Poland) together with another 500 µL LS buffer were added to the remaining debris and subjected to bead-beating in TissueLyser LT (Qiagen, Hilden, Germany). The mechanical lysis was set at 50 oscillations/s for 3 min, according to Salonen et al., 2010 [42]. Samples were again centrifuged, and the bead-beating process was repeated. Then the last lysis mixture was incubated at 95 °C for 5 min to improve cellular degradation. After spinning, the three fractions of supernatants were collected in a single microtube and subjected to purification according to the abovementioned protocol [41] with the Genomic Mini AX Bacteria+ kit (A&A Biotechnology, Gdynia, Poland).

### 4.4. Quantitative Real-Time PCR

A quantitative real-time PCR was performed according to the author’s protocol described earlier [21]. The following genes were used as targets in real-time PCR: the *mcrA* gene encoding methyl-coenzyme M reductase alpha subunit for total methanogenic archaea and the *nifH* gene, which does not encode a functional nitrogenase enzyme for the *Mb. smithii,* the 16S rRNA gene for Methanomassiliicoccales, and the *mtaB* gene encoding coenzyme M methyltransferase for *Ms. stadtmanae*. The specificity of the primers used in this study (Table 4) was checked experimentally by sequencing and using BLAST and modified accordingly. The number of generated amplicons per microorganism was also checked using BLAST and the Ribosomal RNA Database [43]. With the exception of *mtaB*, all target genes occur in a single copy per genome. For *Ms. stadtmanae*, the results of the real-time PCR were divided by four—which is the number of operons per genome in *Ms. stadtmanae* DSM 3091 (acc.: CP000102.1)—in order to achieve the number of cells per gram of feces. Standard curves were generated using decimal dilutions, from approx. 10^0^ to 10^6^ copies per reaction of genomic reference DNAs, which were a linearized plasmid containing an insert of the *mcrA* sequence fragment from GenBank acc. KF214818.1:976-1447, and purified amplicons of *nifH*, 16S rDNA, and *mtaB* generated in the initial screening experiments on the human fecal samples whose sequences most closely aligned to *Methanobrevibacter smithii* strain KB11 (GenBank acc. no.: CP017803.1), uncultured *Methanomassiliicoccus* sp. (GenBank acc. no. LC473299.1), and *Methanosphaera stadtmanae* isolate MGYG-HGUT-02164 (GenBank acc. no. LR698975.1), respectively. The concentrations of the standards were measured with a Quantus fluorometer and the QuantiFluor dsDNA System (Promega Corporation, Madison, WI, USA), and they were further converted into the number of genome copies per µL by using the Science Primer web tool [44].

The real-time PCR mixture included 10 µL of RT HS-PCR Mix SYBR A (A&A Biotechnology, Gdynia, Poland), 0.5 µM primers, approx. 70–100 ng of sample DNA and water to reach a final volume of 20 µL. The thermal conditions—set experimentally in a gradient PCR—are presented in Table 5. In each reaction, the amplification comprised 47 cycles. The real-time PCR results were calculated into the number of cells per gram (dry weight) of the stool sample.

### 4.5. Statistical Analysis 

The statistical analyses were performed in TIBCO Statistica 13.3 (TIBCO Software Inc., Palo Alto, CA, USA). A Shapiro–Wilk test was used to check whether the quantification results were normally distributed. The homogeneity of variance was checked using the Levene’s test. After that, a non-parametric Kruskal–Wallis H test was applied to evaluate the statistical significance of variation among (1) the methanogen groups in relation to the disease type and its activity, (2) the IBD groups in relation to age, PCDAI or PUCAI scores, and FCP. The Spearman’s rank correlation test was used to measure the strength and direction of the methanogenic associations grouped by the disease index activity, calprotectin levels, and age. According to the guidelines for interpretation of Spearman’s rank correlation by Prion and Haerling, 2014 [47], the correlations were considered very strong when the values of Rs were between 0.81 and 1, strong—0.61 and 0.80, moderate—0.41 and 0.60, weak—0.21 and 0.40, and negligible—0 and 0.20. The prevalence of methanogens was analyzed using Fisher’s exact test, whereas the odds ratio analysis was performed in a two-by-two table in the OpenEpi web tool [22].

## 5. Conclusions

Our results showed that methanogenic archaea present in the gut are related to the type of pediatric IBD. This was particularly evident in children with UC, in whom the decrease in methanogens was associated with the presence of the disease. To the best of our knowledge, this study is the first to indicate the possible involvement of *Ms. stadtmanae* in UC, whose increase (in the inactive UC group) was positively correlated with elevated levels of FCP, a known biomarker of mucosal inflammation. Despite the promising results obtained, further studies on a larger scale are needed to assess the involvement of methanogenic archaea in the pathogenesis of IBD in children.

## Figures and Tables

**Figure 1 ijms-25-00673-f001:**
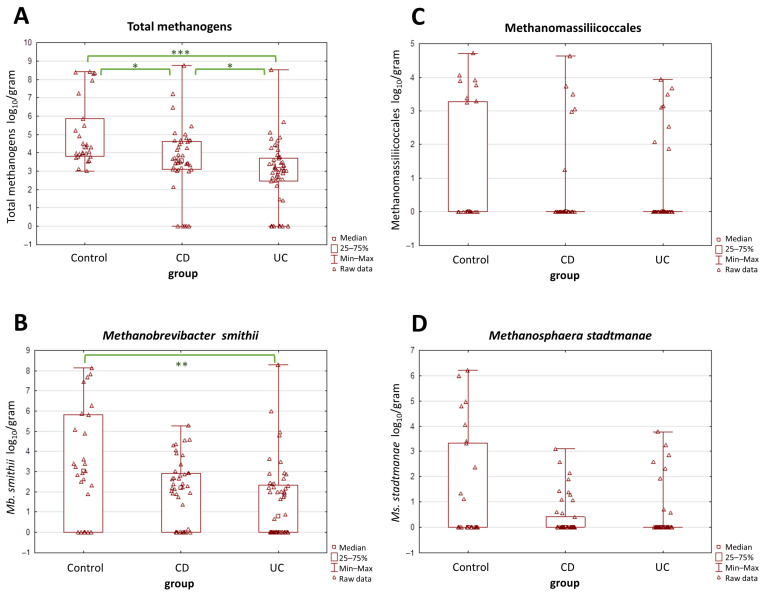
(**A**–**D**) The comparison of methanogen population quantities in stool samples (dry weight) among patients with Crohn’s disease (CD), ulcerative colitis (UC), and the control group. The most pronounced differences were observed in the total population of methanogens (**A**). Values of * *p* < 0.05, ** *p* < 0.01, and *** *p* < 0.001 were regarded as significant. The non-significant results remained unmarked. The statistical analysis was performed using the Kruskal–Wallis H test.

**Figure 2 ijms-25-00673-f002:**
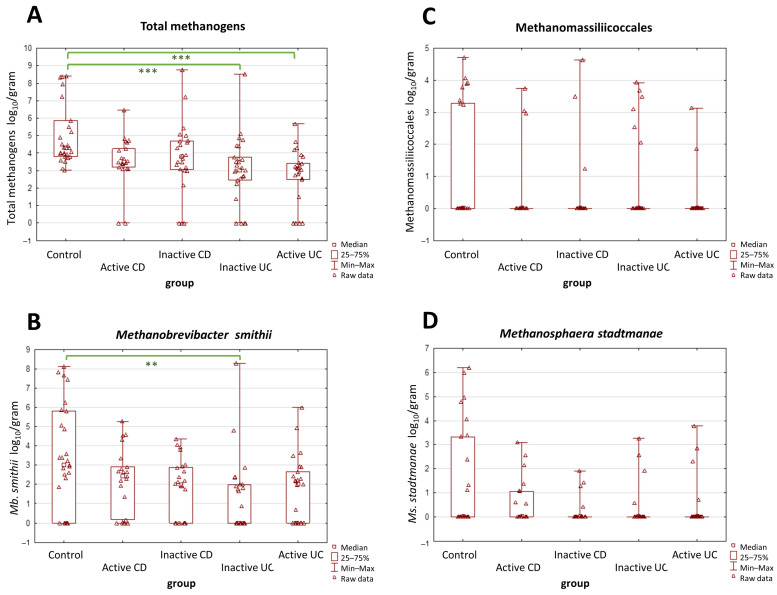
(**A**–**D**) The comparison of methanogen population quantities in the stool samples (dry weight) among patients with active and inactive Crohn’s disease (CD), among patients with active and inactive ulcerative colitis (UC), and the control group. The most pronounced differences were observed in the total population of methanogens (**A**). Values of ** *p* < 0.01 and *** *p* < 0.001 were regarded as significant. The non-significant results remained unmarked. The statistical analysis was performed using the Kruskal–Wallis H test.

**Figure 3 ijms-25-00673-f003:**
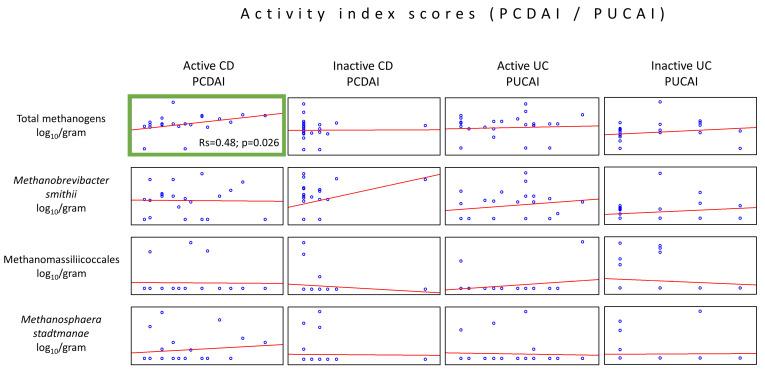
Correlations between the methanogen population quantities and patient disease activity indices (PCDAI or PUCAI) in the stool samples (dry weight) of patients from the IBD groups. The only statistically important correlations were observed in the quantity of the total population of methanogens, which increased with the rise in the PCDAI values (green box).

**Figure 4 ijms-25-00673-f004:**
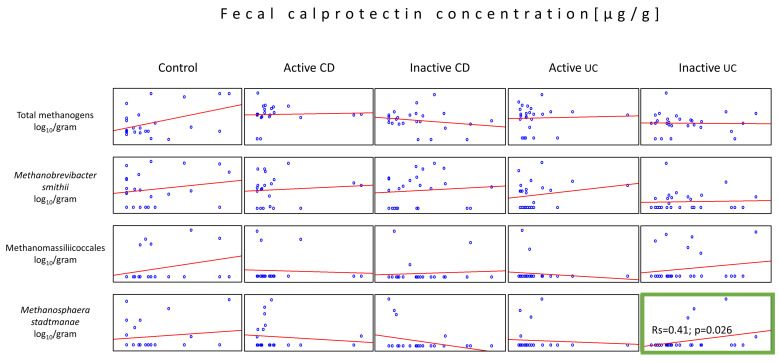
Correlations between the methanogen population quantities and the fecal calprotectin (FCP) levels in the stool samples (dry weight) of patients from all tested groups. The statistically important correlation with respect to FCP was observed in the population of *Ms. stadtmanae* in the inactive UC group (green box).

**Figure 5 ijms-25-00673-f005:**
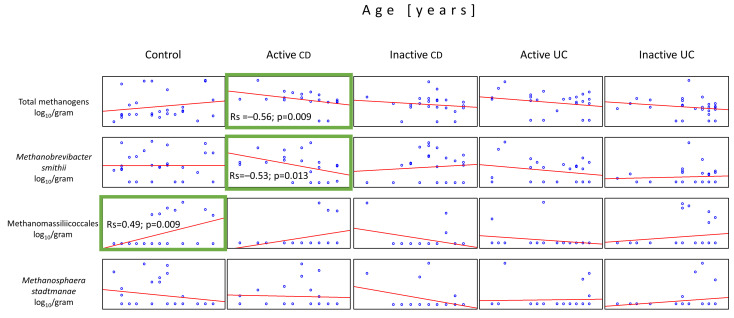
Correlations between the quantities of the methanogen populations and the patient’s age in the stool samples (dry weight) of patients from all tested groups. The only statistically important correlations with respect to the patient’s age were observed in populations of Methanomassiliicoccales in the controls and total methanogens and *Mb. smithii* in the active CD group (green boxes).

**Table 1 ijms-25-00673-t001:** The characteristics of the subjects.

Group	Mean Age [Years] ± SD (Mdn)	Female to Male Ratio	Mean PCDAI ± SD (Mdn)	Mean PUCAI ± SD (Mdn)	Mean FCP [µg/g] ± SD (Mdn)
CD (n = 45)	14.2 ± 3.1 (15.0) *	25:20	10.0 ± 13.5 (5.0)	n/a	781.6 ± 1800.7 (119.0) *
Active CD (n = 21)	13.9 ± 3.4 (15.0) *	15:6	17.5 ± 14.7 (15.0) #	n/a	1601.8 ± 2409.4 (699.0) *#
Inactive CD (n = 24)	14.6 ± 2.8 (15.0) *	10:14	3.4 ± 7.8 (0.0)	n/a	64.0 ± 61.0 (59.5)
UC (n = 52)	13.0 ± 4.7 (14.0) *	27:25	n/a	15.2 ± 20.3 (5.0)	396.9 ± 810.3 (100.5) *
Active UC (n = 25)	12.2 ± 5.0 (13.0)	12:13	n/a	27.6 ± 23.3 (25.0) #	758.5 ± 1062.9 (403.0) *#
Inactive UC (n = 27)	13.8 ± 4.3 (15.0) *	15:12	n/a	3.7 ± 4.9 (0.0)	62.0 ± 54.9 (38.0)
Controls (n = 27)	10.0 ± 4.0 (10.0)	14:13	n/a	n/a	19.3 ± 24.1 (10.0)

n/a—not applicable; SD—standard deviation; Mdn—medians presented in parentheses (); PCDAI—Pediatric Crohn’s Disease Activity Index; PUCAI—Pediatric Ulcerative Colitis Activity Index; FCP—fecal calprotectin concentration; CD—Crohn’s disease; UC—ulcerative colitis; * statistical significance (*p* < 0.05) compared to control; # statistical significance (*p* < 0.05) compared to the inactive group of the same disease type. The statistical analysis was performed using the Kruskal–Wallis H test.

**Table 2 ijms-25-00673-t002:** The prevalence of the methanogens [%] in the analyzed pediatric groups of the active and inactive types of IBD and controls.

Group	No. of Positive/Total No. of Tested Samples and Percentage Values [%]			
	Total Methanogens	*Mb. smithii*	*Ms. stadtmanae*	Methanomasiilicoccales
CD (n = 45)	40/45 [88.9%]	31/45 [68.9%]	12/45 [26.7%]	6/45 [13.3%]
Active CD (n = 21)	19/21 [90.5%]	16/21 [76.2%]	8/21 [38.1%]	3/21 [14.3%]
Inactive CD (n = 24)	21/24 [87.5%]	15/24 [62.5%]	4//24 [16.7%]	3/24 [12.5%]
UC (n = 52)	43/52 [82.7%] *	27/52 [51.9%]	8/52 [15.4%] *	8/52 [15.4%]
Active UC (n = 25)	20/25 [80.0%] *	15/25 [60.0%]	4/25 [16.0%]	2/25 [8.0%]
Inactive UC (n = 27)	23/27 [85.2%]	12/27 [44.4%]	4/27 [14.8%]	6/27 [22.2%]
Controls (n = 27)	27/27 [100%]	20/27 [74.1%]	10/27 [37.0%]	8/27 [29.6%]

The results of the percentage of children whose stool samples tested positive for total methanogens, *Mb. smithii*, *Ms. stadtmanae*, and Methanomassiliicoccales. The prevalence of methanogenic archaea was measured by real-time PCR [21]. * statistical significance (*p* < 0.05) compared to control; CD—Crohn’s disease; UC—ulcerative colitis. Statistical analysis was performed using Fisher’s exact test.

**Table 3 ijms-25-00673-t003:** The odds of detection of methanogens in the analyzed pediatric groups of the active and inactive types of IBD compared to controls.

Group	Odds Ratios [95% Confidence Intervals] Compared to Control Group			
	Total Methanogens *	*Mb. smithii*	*Ms. stadtmanae*	Methanomasiilicoccales
CD (n = 45)	0	0.78 [0.27, 2.25]	0.62 [0.22, 1.72]	0.37 [0.11, 1.20]
Active CD (n = 21)	0	1.12 [0.30, 4.20]	1.05 [0.32, 3.40]	0.40 [0.09, 1.73]
Inactive CD (n = 24)	0	0.58 [0.18, 1.92]	0.34 [0.09, 1.28]	0.34 [0.08, 1.47]
UC (n = 52)	0	0.38 [0.14, 1.05]	0.31 [0.10, 0.91]	0.43 [0.14, 1.32]
Active UC (n = 25)	0	0.53 [0.16, 1.70]	0.32 [0.09, 1.22]	0.21 [0.04, 1.09]
Inactive UC (n = 27)	0	0.28 [0.09, 0.88]	0.30 [0.08, 1.11]	0.68 [0.20, 2.31]

* none of the control subjects tested negative for total methanogens; therefore, the results equal zero; CD—Crohn’s disease; UC—ulcerative colitis. The odds ratio analysis was performed in the OpenEpi web tool [22].

**Table 4 ijms-25-00673-t004:** Primers used for the determination of total methanogens and their three subgroups.

Microorganism	Target Gene	Forward/Reverse Primer 5′–3′ Sequence *	Amplicon Length [bp]	Reference
Total methanogenic archaea	*mcrA*	CTTGAARMTCACTTCGGTGGWTC/CGTTCATBGCGTAGTTVGGRTAGT	Approx. 270	[21]
*Methanobrevibacter smithii*	*nifH*	AACAGAAAACCCAGTGAAGAGGATA/ACGTAAAGGCACTGAAAAACCTCC	222	Modified [45]
Methanomassiliicoccales	16S rDNA	GGGGTAGGGGTAAAATCCTGTAATCC/AACAACTTCTCTCCGGCACTGG	194	Modified [46]
*Methanosphaera stadtmanae*	*mtaB*	GTAGTTCCTAACATCAAAGTAGCTCC/TCCTCTAAGACCGTTTTCTTCTTCTCTCA	300	Modified [16]

* original oligo sequence from the cited publication is underlined.

**Table 5 ijms-25-00673-t005:** Temperature settings used for the absolute quantification of each group of methanogenic archaea.

Real-Time PCR Step	Total Methanogens	*Mb. smithii*	Methanomassiliicoccales	*Ms. stadtmanae*
Initial Denaturation	95 °C—5 min			
Denaturation	94 °C—20 s	94 °C—20 s	94 °C—20 s	94 °C—20 s
Annealing	60 °C—20 s	66 °C—20 s	70 °C—20 s	68 °C—20 s
Elongation	72 °C—20 s	72 °C—20 s	72 °C—20 s	72 °C—25 s
Signal acquisition *	81 °C—20 s + Acq	82 °C—20 s + Acq	87 °C—20 s + Acq	81 °C—20 s + Acq
Melt analysis *	95 °C—5 s, then 60 °C—1 min, and 95 °C—continuous Acq with ramp rate 0.11 °C/s			

* Acq—acquisition of fluorescence signal.

## Data Availability

The data presented in this study are available upon request from the corresponding author. The data are not publicly available due to privacy protections.

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
