# Peer review of "Methanogenic Archaea in the Pediatric Inflammatory Bowel Disease in Relation to Disease Type and Activity"

_ijms, 2024, doi:10.3390/ijms25010673_

Round 1
Reviewer 1 Report
Comments and Suggestions for Authors
This is an interesting manuscript, but several changes should be made before it can be published, those addition will improve this manuscript:
Minor issues
1) When talking about IBD, mention the prevalence in other places, for example in America, or the world.
2) Briefly describe IBD. What consequences does IBD have? Keep in mind that not all readers are familiar with the disease.
3) Explain in a paragraph what archaea are, you have to keep in mind that the article can be read by a reader, for example, an internal medicine doctor or a pediatrician, who is unfamiliar with this. For example, you can comment that the first ones described were the Extremophiles, and later it was seen that they could be in the digestive system, etc.
The tables should be understood without having to read the text, so at the bottom of table 2 it should be explained what CD, UC, etc. are. (The authors have explained the acronyms in the figures but not in the tables.)
Major issues
4) Use the data from table 2, for creating an additional new table but with the Odds ratio and its 95% confidence intervals. You can do this with the OpenEpi program https://www.openepi.com/TwobyTwo/TwobyTwo.htm
Author Response
Dear Reviewer,
thank you for your thorough review and valid remarks. We have made the following changes:
Remark 1: When talking about IBD, mention the prevalence in other places, for example in America, or the world.
Response: This information has been added in the Introduction section (lines 40-44).
Remark 2: Briefly describe IBD. What consequences does IBD have? Keep in mind that not all readers are familiar with the disease.
Response: This information has been added in the Introduction section (lines 33-37).
Remark 3: Explain in a paragraph what archaea are, you have to keep in mind that the article can be read by a reader, for example, an internal medicine doctor or a pediatrician, who is unfamiliar with this. For example, you can comment that the first ones described were the Extremophiles, and later it was seen that they could be in the digestive system, etc.
Response: This information has been added in the Introduction section (lines 50-54).
Remark 4: The tables should be understood without having to read the text, so at the bottom of table 2 it should be explained what CD, UC, etc. are. (The authors have explained the acronyms in the figures but not in the tables.)
Response: All acronyms have been explained in the tables 1, 2 and 3.
Remark 5: Use the data from table 2, for creating an additional new table but with the Odds ratio and its 95% confidence intervals. You can do this with the OpenEpi program https://www.openepi.com/TwobyTwo/TwobyTwo.htm
Response: Table 3 has been created (with the odds ratio and 95% confidence intervals), and a short referral to this table has been added to the main text (lines 108-110 and 401-402).
Reviewer 2 Report
Comments and Suggestions for Authors
In the study, Cisek et al. investigated the possible association between IBD and methanogenic archaea in children. Specifically, three subgroups of methanogenic archaea, two groups of patients (with CD or UC), and PCDIA, PUCAI, and FCP were selected for comparison. In general, the study design and the data presented are of acceptable quality. Below are some comments for consideration.
1. Abstract, line 21, spell out "PCDAI" and "PUCAI".
2. Table 1, column "Mean FCP [μg/g] ± SD (Mdn)". For both CD and UC groups, some of the SD values are much larger than the corresponding mean values. Did the authors perform an outlier analysis and remove unusually large values? Are the observations here consistent with previously reported values, especially those obtained from adult patients?
3. Lines 78-79: Remove the text.
4. Figures 1-3: It appears that if the total methanogenic archaea were used for comparison, there would be some difference between the control group and the patient groups, whereas no such difference was observed if the individual subgroups of methanogenic archaea were used for comparison. Is it possible that more subgroups of methanogenic archaea should have been included in the study?
5. Were any sex-related patterns observed?
6. In the discussion section, recommend that the observations and findings of the study be compared with those of adult patients.
7. Based on the results of this study, would the authors be able to suggest any treatments regarding IBD in young patients?
Comments on the Quality of English LanguageMinor editing of English language required.
Author Response
Dear Reviewer,
thank you for your thorough review and valid remarks. We have made the following changes:
Remark 1: Abstract, line 21, spell out "PCDAI" and "PUCAI".
Response: The acronyms have been explained.
Remark 2: Table 1, column "Mean FCP [μg/g] ± SD (Mdn)". For both CD and UC groups, some of the SD values are much larger than the corresponding mean values. Did the authors perform an outlier analysis and remove unusually large values? Are the observations here consistent with previously reported values, especially those obtained from adult patients?
Response: High FCP values are related to disease activity – the higher FCP concentration the more active the disease, and the activity of disease was at the core of the experimental setup. Moreover, FCP data were acquired in patients, who are naturally very diverse among each other as disease activity is also different. For these reasons we did not perform an outlier analysis, and we could not have excluded any patients.
Based on the literature reports, a wide range of FCP values in IBD patients is common. Konikoff & Denson (2006) published a review of FCP values reported in the literature [PMID: 16775498]. There the widest range of FCP values in the adults with IBD was between 27 to 6850 with median 496 µg/g, whereas in the children with IBD – between 40 to 8575 with median 237 µg/g. In the light of these findings our results seem consistent.
Remark 3: Lines 78-79: Remove the text.
Response: The text has been removed.
Remark 4: Figures 1-3: It appears that if the total methanogenic archaea were used for comparison, there would be some difference between the control group and the patient groups, whereas no such difference was observed if the individual subgroups of methanogenic archaea were used for comparison. Is it possible that more subgroups of methanogenic archaea should have been included in the study?
Response: There is a lot of things we still don’t know about methanogens. We do know for sure that the three studied subgroups are the most prevalent and abundant in the human gut. There are some other methanogens occasionally reported as well, but mostly in adults and in far lower quantities. It is possible that our pediatric patients might have been colonized by other unstudied taxa, and it cannot be ruled out that these taxa might have been somehow related with the incidence of IBD in our patients. However, to date not a single archaeal species has been identified as clearly pathogenic. Therefore this question remains open until we learn more about the diversity of methanogenic archaea in the children’s intestines. Due to the importance of the issue raised, we decided to add this information to the Discussion section (lines 288-294).
Remark 5: Were any sex-related patterns observed?
Response: No, there was no sex-related pattern of any kind.
Remark 6: In the discussion section, recommend that the observations and findings of the study be compared with those of adult patients.
Response: This information has been added in the discussion section (lines 239-240)
Remark 7: Based on the results of this study, would the authors be able to suggest any treatments regarding IBD in young patients?
Response: Based on our results we cannot suggest anything in particular regarding the treatment of pediatric IBD. There was no correlation between the treatment type and any of the tested methanogen group. It was probably due to a small number of observations per therapy, and it is possible that some relations may have been omitted in our work. Importantly, even though there were methanogen-negative samples, they never grouped solely in a particular drug group. Therefore, none of the prescribed therapeutic agent could be associated with a complete elimination of methanogens. This is the only conclusion treatment-wise that comes from our study, which has already been discussed in the “Limitation and strength of the study” of the Discussion section (lines 280-284).
Round 2
Reviewer 2 Report
Comments and Suggestions for Authors
The authors have addressed my comments and revised the original manuscript accordingly. Therefore, I would like to recommend the current version for publication.